# Non-Coding RNAs and Prediction of Preeclampsia in the First Trimester of Pregnancy

**DOI:** 10.3390/cells11152428

**Published:** 2022-08-05

**Authors:** Manabu Ogoyama, Hironori Takahashi, Hirotada Suzuki, Akihide Ohkuchi, Hiroyuki Fujiwara, Toshihiro Takizawa

**Affiliations:** 1Department of Obstetrics and Gynecology, Jichi Medical University, 3311-1 Yakushiji, Shimotsuke 329-0498, Japan; 2Department of Molecular Medicine and Anatomy, Nippon Medical School, 1-1-5 Sendagi, Tokyo 113-8602, Japan

**Keywords:** preeclampsia, non-coding RNA, placentation, first trimester, prediction

## Abstract

Preeclampsia (PE) is a major cause of maternal and perinatal morbidity and mortality. The only fundamental treatment for PE is the termination of pregnancy. Therefore, not only severe maternal complications but also perinatal complications due to immaturity of the infant associated with early delivery are serious issues. The treatment and prevention of preterm onset preeclampsia (POPE) are challenging. In 2017, the ASPRE trial showed that a low oral dose of aspirin administered to POPE high-risk women in early pregnancy reduced POPE by 62%. A prediction algorithm at 11–13 weeks of gestation identifies POPE with 75% sensitivity when the false positive rate is set at 10%. New biomarkers to increase the accuracy of the prediction model for POPE high-risk women in early pregnancy are needed. In this review, we focused on non-coding RNAs (ncRNAs) as potential biomarkers for the prediction of POPE. Highly expressed ncRNAs in the placenta in early pregnancy may play crucial roles in placentation. Furthermore, placenta-specific ncRNAs have been detected in maternal blood. In this review, we summarized ncRNAs that were highly expressed in the primary human placenta in early pregnancy. We also presented highly expressed ncRNAs in the placenta that were associated with or predictive of the development of PE in an expression analysis of maternal blood during the first trimester of pregnancy. These previous studies showed that the chromosome 19 microRNA (miRNA) -derived miRNAs (e.g., *miR-517-5p*, *miR-518b*, and *miR-520h*), the hypoxia-inducible miRNA (*miR-210*), and long non-coding RNA *H19*, were not only highly expressed in the early placenta but were also significantly up-regulated in the blood at early gestation in pregnant women who later developed PE. These maternal circulating ncRNAs in early pregnancy are expected to be possible biomarkers for POPE.

## 1. Introduction

### 1.1. Preeclampsia

Preeclampsia (PE), which occurs in 2 to 8% of all pregnancies, is a major cause of maternal and perinatal morbidity and mortality [1,2,3]. Although the definition of PE slightly differs between regions and countries, it is a condition of hypertension that develops after 20 weeks of gestation and is complicated by organ and uteroplacental dysfunctions [4,5]. The only fundamental treatment for PE is the termination of pregnancy. Therefore, not only severe maternal complications, including hepatic/renal failure and cerebral hemorrhage, but also perinatal complications due to immaturity of the infant associated with early delivery (e.g., intraventricular hemorrhage, cerebral palsy, respiratory distress syndrome, chronic lung disease, necrotizing enterocolitis, sensory disorders, and susceptibility to infection) are serious issues in pregnant women with preterm onset PE (POPE) (onset at <37 weeks of gestation) [6,7]. The treatment and prevention of POPE are challenging in perinatology.

### 1.2. Pathogenesis of PE

The two-step theory is the main mechanism underlying the development of PE. In early pregnancy, extravillous trophoblasts (EVT) differentiated from proximal cell columns derived from cytotrophoblasts invade the maternal decidua and uterine myometrium [1,2,8]. In this process, the vascular endothelial cells of the uterine spiral arteries are replaced by invading EVT, which dilates these arteries. This remodeling of the uterine spiral arteries contributes to the supply of blood rich in oxygen and nutrients to the intervillous space of the placenta, resulting in normal placentation. In pregnant women with PE, a dysfunction in EVT invasion in early pregnancy may cause placental hypoxia and ischemia (first step), triggering the release of anti-angiogenic factors, such as soluble fms-like tyrosine kinase-1 (sFlt-1) and soluble endoglin, into the maternal blood. It may also cause systemic endothelial disorders, resulting in maternal hypertension and fetoplacental dysfunction (second step) [1,2,9]. Therefore, EVT invasion in early pregnancy is indispensable for normal placentation and pregnancy.

### 1.3. Prediction and Prevention of PE

In 2017, the ASPRE (Combined Multimarker Screening and Randomized Patient Treatment with Aspirin for Evidence-Based Preeclampsia Prevention) trial showed that a low oral dose of aspirin (LDA: 150 mg/day from 11–14 to 36 weeks of gestation) reduced POPE by 62% and early onset PE (EOPE) (onset at <34 weeks of gestation) by 82% in pregnant women who were identified as being at high risk of POPE at 11–13 weeks of gestation; the reduction in EOPE was not significant [10]. Therefore, it is important to effectively identify pregnant women at high risk of POPE and EOPE in early pregnancy and administer LDA medication at less than 16 weeks of gestation in order to reduce the maternal and neonatal complications associated with early delivery due to POPE. In the ASPRE trial, women at high risk of POPE were identified by the combination method using maternal factors, mean arterial pressure (MAP), the uterine artery pulsatility index (PI), and maternal serum pregnancy-associated plasma protein A (PAPP-A), and placental growth factor (PlGF) [11]. This prediction model was established by the Fetal Medicine Foundation, and a revised algorithm using maternal characteristics and medical history, MAP, uterine artery PI, and serum PlGF identifies preterm PE with 75% sensitivity and EOPE with 90% sensitivity when the false positive rate is set at 10% [12]. It was also shown to be equally accurate in other cohort studies [13,14,15,16,17,18,19,20]. We currently use this prediction model online (https://fetalmedicine.org/research/assess/preeclampsia/first-trimester [accessed 1 July 2022]). However, 25% of pregnant women who are originally eligible for LDA medication will be missed when this prediction model is used. Therefore, new biomarkers to increase the accuracy of the prediction model for POPE high-risk women in early pregnancy are needed (Figure 1). In addition to serum PlGF, sFlt1, PAPP-A, and galectin 13 (LGALS13, known as PP13), candidate factors as components of the FMF prediction model, the following PE biomarkers in the first trimester have been verified: (1) Protein markers, such as ADAM metallopeptidase domain 12 (ADAM12), a type of matrix metalloproteinase, and P-selection, a cell surface adhesion molecule involved in inflammation [21]; (2) various metabolic concentrations (e.g., taurine and asparagine) obtained by metabolomics profiling [22]; and (3) cell-free fetal DNA, which is released from placental cells during apoptosis and is detectable in maternal blood from 9 weeks of gestation, were shown to be predictive of PE and EOPE in the first trimester [21,23,24]. These biomarkers have also been evaluated for their predictive ability in combination with MAP and uterine artery PI, which are used in FMF predictive models; however, their accuracy has not been sufficient to improve the predictive ability of the FMF model.

### 1.4. Non-Coding RNA

In this review, we focused on non-coding RNAs (ncRNAs) as potential biomarkers for the prediction of POPE. ncRNAs without coding potential have also attracted attention for their involvement in the regulation of gene expression. More than 75% of the human genome is transcribed into ncRNAs, in contrast to 1–2% into protein-coding messenger RNAs (mRNAs) [25]. ncRNAs are subdivided into two groups based on their sizes: small ncRNAs (<200 bp) and long ncRNAs (lncRNAs: >200 bp). Small ncRNAs, including microRNA (miRNA), piwi-interacting RNA, small interfering RNA, and small nucleolar RNA, mainly function as negative regulators of gene expression [25,26]. miRNAs with lengths of 22–25 bp have approximately 2700 species in humans, and suppress their target gene expression by destabilizing or impairing translation through binding their 3′ untranslated region (3′UTR) [27]. lncRNAs may play important roles in biological processes via their various functions, including signals (regulating transcription and translation), antisense decoys (interfering complementary mRNAs and miRNAs), transcriptional interference, scaffolds (forming functional complexes via binding proteins and other RNAs), and generating miRNAs [28,29,30,31]. The expression and effects of lncRNAs are dependent on the cell type, tissue type, and time [32]. There are several databases that predict target genes based on the sequences of miRNAs and mRNAs [33,34], and various mutual interactions between miRNAs and their target genes have been experimentally demonstrated. However, many mutual interactions that have yet to be identified are speculated to exist. Although the number of lncRNAs with known functions has increased in the past few decades, the functions of the majority of lncRNAs, with more than 19,000 species in humans, remain unclear.

### 1.5. Possible ncRNAs as Predictive Markers of PE

Recent studies reported ncRNAs that are highly expressed in the placenta. Highly expressed ncRNAs in the placenta in early pregnancy, particularly trophoblasts, including EVT, may play crucial roles in EVT invasion. Placenta-specific ncRNAs have also been detected in maternal blood because chorionic villi are in contact with maternal blood at the intervillous space [35]. Therefore, pregnant women who develop PE may have abnormal expression levels of these ncRNAs in the blood in early pregnancy. Evaluations of the expression levels of maternal circulating placenta-specific ncRNAs in early pregnancy are expected to predict the onset of POPE. Blood sampling is less invasive for the mother. Although many ncRNAs have been reported as candidate predictors of PE, most have been analyzed for their expression in the placenta and maternal blood of post-onset PE and week-adjusted control pregnant women [36,37,38,39,40,41]. However, ncRNA expression levels in the placenta and maternal blood after the onset of PE may reflect systemic vascular endothelial dysfunction rather than the pathogenesis or cause of PE. Therefore, evaluations of ncRNA expression levels in the placenta and maternal blood in early pregnancy are essential for the development of a prediction model for pregnant women at high risk of POPE/EOPE. However, few studies have performed expression and functional analyses of ncRNAs using placenta and blood samples obtained in early pregnancy.

In this review, we summarized ncRNAs that were highly expressed in the primary human placenta in early pregnancy and may play an important role in placentation based on functional analyses. We also presented highly expressed ncRNAs in the placenta that were reported to be associated with or predictive of the development of PE by expression analyses of maternal blood during the first trimester of pregnancy.

## 2. Highly Expressed ncRNAs in the First Trimester Placenta

The number of studies on ncRNAs highly expressed in the early placenta has been increasing. Conventional methods for gene expression analyses of the placenta in early pregnancy mainly focused on quantification by real-time PCR (RT-PCR) and localization evaluations by in situ hybridization (ISH). Furthermore, ncRNA expression levels were examined within more restricted structural regions of the placenta by laser microdissection (LMD) [42,43]. In recent years, many studies have comprehensively quantified the expression levels of ncRNAs in the human early placenta and trophoblasts by RNA sequencing and ranked their expression levels [44,45,46,47]; some also performed an in silico analysis based on the expression levels of ncRNAs in the early placenta to study their functions and related pathways in early placentation. Gene expression levels were evaluated at the cellular level in the early placenta by single-cell RNA sequencing; however, ncRNA expression levels were not subjected to a similar analysis [48,49,50]. Table 1 summarizes the findings obtained by comprehensive gene expression profiling of the early placenta using microarray and RNA sequencing analyses [44,45,46,47,51,52]. Table 2 summarizes findings from gene expression evaluations and the validation of individual ncRNAs in the human primary early placenta and trophoblasts [42,43,53,54,55,56,57,58,59,60,61,62]. ncRNAs that are highly expressed in the human early placenta, as shown in Table 1 and Table 2, may play important roles in EVT invasion and placentation and have potential as biomarkers for the prediction of PE.

### 2.1. miRNAs

#### 2.1.1. Placenta-Specific miRNA Clusters

Previous RNA sequencing and microarray analyses using human placentas showed the specific expression of miRNAs in the placenta. They form clusters on some chromosomes, representative of which are the chromosome 19 miRNA cluster (C19MC), chromosome 14 miRNA cluster (C14MC), and *miR-371-3* cluster [63]. C19MC is an approximately 100-kb-long imprinted gene cluster located on 19q13.41 and consists of 46 miRNAs expressed only from the paternally inherited chromosome [63,64]. Using microarray analysis and RT-PCR validation, Morales–Prieto et al. showed that C19MC miRNAs were highly expressed in the early placenta, with further increases being observed with the progression of pregnancy [51]. The *miR-371-3* cluster is located approximately 1000 bp downstream of C19MC and consists of 3 miRNAs: *miR-371a-3p*, *miR-372*, and *miR-373-3p* [63]. C14MC is located on an imprinted *DLK-DIO3* region on 14q32, which contains 52 maternally inherited miRNAs [51]. In contrast to C19MC miRNAs, C14MC miRNAs are generally down-regulated or invariant throughout pregnancy. Gonzalez et al. demonstrated using a miRNA sequencing analysis of human placentas that the expression of C19MC miRNAs increased while that of C14MC miRNAs decreased with gestation [46]. On the other hand, Smith et al. performed a miRNA sequencing analysis of placentas at 6–23 weeks of gestation, and a comparison of the miRNA expression status between 6–10 and 11–23 weeks of gestation showed that the expression of C19MC miRNAs was more repressed while that of C14MC miRNAs was up-regulated at 11–23 weeks of gestation [47]. Gene expression clustering of human trophoblast and choriocarcinoma cell lines based on microarray analyses revealed that the gene expression status of C19MC miRNAs in human trophoblasts was similar to that of the EVT cell line (HTR-SVneo), while that of C14MC miRNAs was similar to that of the choriocarcinoma cell line (JEG3) [51]. C19MC and C14MC miRNAs have been shown to play a role in not only EVT invasion but also tumorigenesis through gene expression networks via the repression of their various target genes [43,63]. In addition, they exert other effects, such as antiviral effects on trophoblast cells; however, the underlying mechanisms remain unclear [65,66].

Placental villi are in direct contact with maternal blood in the intervillous space, and, thus, the expression status of placenta-specific miRNAs in the maternal circulation has been evaluated. Luo et al. performed a small RNA library sequencing analysis of the placenta and plasma of early pregnant women and listed the miRNAs that were highly expressed in each sample group [44]. C19MC miRNAs accounted for 30–40% of miRNAs, and the most highly expressed miRNA in both the placenta and plasma was miR-21 on chromosome 17, which is not in the placenta-specific miRNA cluster. An expression analysis of human adult organs by RT-PCR showed that C19MC miRNAs, such as *miR-517a*, *miR-517b*, *miR-518b*, *miR-519a*, and *miR-512-3p*, were exclusively expressed in the placenta, whereas *miR-21* was ubiquitously expressed. In addition, RT-PCR analysis of maternal plasma 1 day before and 3 days after delivery revealed that the expression of C19MCs markedly decreased after delivery, indicating that C19MC miRNAs were expressed exclusively during pregnancy [44]. Miura et al. also examined the miRNA profile of maternal plasma [52]. They performed a microarray analysis using RNAs obtained from human early and late placentas and plasma and extracted 82 miRNAs that were expressed at more than 100-fold higher levels in the placenta than in plasma. Among the miRNAs examined, the expression levels of 24 decreased after delivery and were reported as “pregnancy-associated miRNAs”. It is important to note that 66.7 and 20.8% of these miRNAs were derived from C19MC and C14MC, respectively. We previously identified highly expressed ncRNAs in EVT and chorionic villous trophoblasts (CVT) using an RNA sequencing analysis of EVT and CVT isolated from human early placentas [45]. An interesting feature in the expression pattern of miRNAs in trophoblast cells was that the expression of all miRNAs on chromosome 14 was lower in EVT than in CVT. This finding indicated that the repression of C14MC miRNA expression in EVT is involved in EVT invasion in early pregnancy. Collectively, these findings suggest that C19MC and C14MC miRNAs play crucial roles in placentation.

The miRNA expression of placenta-specific miRNA clusters has been visually confirmed by ISH using human early placental samples, and their expression has been validated by RT-PCR with or without LMD. Wang et al. evaluated the expression of 8 C19MC miRNAs in the placentas of elective termination cases at 6–9 weeks gestation by RT-PCR and ISH [53]. Although their expression levels varied depending on the site and weight of the placentas, ISH showed that the expression of *miR-517b* and *miR-519a* was concentrated in trophoblast cells. Mong et al. focused on EVT invasion in early pregnancy and confirmed the expression distribution of the C19MC miRNA, *miR-517a/c*, in detail by ISH using early placentas at the site at which VT adheres to the maternal decidua and differentiates from the proximal cell columns of CVT to EVT [54]. *miR-517a/c* staining was the most intense in syncytiotrophoblasts (SCT), CVT, and proximal cell columns but gradually decreased as cells differentiated into EVT. In addition, when induced pluripotent stem cells differentiated into EVT-like cells under hypoxic conditions, the expression of C19MC miRNAs decreased, while that of epithelial-to-mesenchymal transition (EMT)-related genes, including *CDH2* and *SERPINE1*, increased. These findings suggest that C19MC miRNAs suppress EVT invasion, namely, the repression of C19MC miRNA expression in EVT promotes EVT invasion. Takahashi et al. and Xie et al. also demonstrated that the expression of *miR-520c-3p* and *miR-519d* on C19MC was lower in EVT than in villous trophoblasts (VT) by RT-PCR following LMD [42,43]. Furthermore, they showed that both miRNAs exhibited the ability to suppress EVT invasion. *miR-520-3p* inhibited cell invasion by directly suppressing *CD44*, a cell-surface receptor for hyaluronic acid, and EVT invasion was enhanced when *miR-520-3p* decreased [42]. *miR-519d* was also shown to indirectly repress cell invasion-promoting genes, such as *CXCL6* and *FOXL2* [43]. These findings describe a model of C19MC miRNA-mediated EVT invasion, in which EVT acquires the ability to invade the maternal decidua and uterine myometrium by decreasing the expression of C19MC miRNAs, which may suppress cell invasion. Flor et al. focused on the epigenetic regulation of C19MC and reported that the methylation of CpG islands in the promotor region upstream of C19MC in the paternal allele was broadly suppressed in human early placentas [55]. Abnormal methylation in the early placenta and chorionic villi may also influence EVT invasion and placentation; however, further studies are needed.

Based on these findings, it may be possible to detect a change in the expression of C19MC miRNAs in the early trophoblasts and peripheral blood of women who later develop PE in early pregnancy because they show insufficient EVT invasion. C19MC miRNAs have potential as biomarkers for PE. Further evaluations of other placenta-specific miRNAs for the prediction of PE are needed.

#### 2.1.2. Hypoxia-Responsive miRNA: miR-210

EVT invasion into the maternal decidua and uterine myometrium has been detected at 9–20 weeks of gestation when the placenta is under hypoxic conditions [1]. Therefore, the expression level of *miR-210*, a hypoxia-inducible miRNA in the human early placenta, is expected to be associated with the invasive ability of EVT, and several researchers have focused on the relationship between *miR-210* and EVT invasion. Hayder et al. compared the expression level of *miR-210-3p* among first, second, preterm, and term human placentas by RT-PCR and found that its expression decreased as gestational weeks increased [56]. The expression level of *miR-210-3p* was the highest in first trimester placentas. In addition, a functional analysis of EVT cells isolated from the human placenta at 6–9 weeks of gestation showed that *miR-210-3p* significantly inhibited EVT outgrowth, migration, and invasion. As one of the underlying mechanisms, they demonstrated the direct inhibitory effects of *miR-210-3p* on *CDX2*, a major transcription factor for the trophectoderm lineage and maintenance of trophoblast self-renewal [56]. Wang et al. also showed that *miR-210* was highly expressed in VT cells by ISH using human early placentas [57,58]. The expression of *miR-210* was attenuated as VT differentiated into EVT [58]. Furthermore, the overexpression of *miR-210* in EVT cell lines suppressed their invasion and migration with the attenuation of EMT markers, while the opposite was observed when *miR-210* expression was suppressed [58]. These findings suggest that hypoxia-inducible *miR-210* is highly expressed in the human primary placenta in early pregnancy. However, its expression in EVT is attenuated, which appears to promote the invasive ability of EVT. This mechanism has not yet been elucidated in detail, and thus, further investigations are needed. The relationship between *miR-210* expression in the peripheral blood of early pregnant women and the development of PE has been examined and will be discussed in a later paragraph.

#### 2.1.3. miRNAs That May Be Involved in Early Placentation Based on an Expression Analysis Using Human Early Placentas

Previous studies investigated ncRNAs that may be involved in early placentation based on comparative gene expression analyses using human placentas, such as PE vs. non-PE and early vs. late pregnancy samples. Gu et al. performed microarrays using RNA extracted from early (6–8 w) and late (30–38 w) human placentas. They validated 3 up-regulated miRNAs (*miR-371-5p*, *miR-17-3p*, and *miR-708-5p*) and 2 down-regulated miRNAs (*miR-125b-5p* and *miR-139-5p*) in early placentas among 191 miRNAs with significantly altered expression levels (fold change: FC ≥ 2.0) by RT-PCR and ISH [59]. Among the 97 miRNAs with significantly higher expression levels in early placentas, 19 were C19MC miRNAs, 5 were C14MC miRNAs, and all 3 were *miR-371* cluster miRNAs. On the other hand, 8 *let-7* family miRNAs, which function as tumor suppressors by inhibiting their target genes [67,68], showed significantly lower expression levels in early placentas than in late placentas. Singh et al. retrospectively analyzed tissues obtained from pregnant women at 11–13 weeks of gestation by chorionic villous sampling (CVS) and revealed using microarrays that the expression levels of 9 miRNAs significantly differed between 4 pregnant women who later developed severe PE and 4 non-pregnant women [60]. The RT-PCR validation of miR-202-3p, which showed the largest expression change, demonstrated that its expression was 7-fold higher in placentas that later developed severe PE. Other tumorigenic studies reported that *miR-202-3p* suppressed cell proliferation and invasion [69,70,71], suggesting its involvement in the pathogenesis of PE (EVT invasion failure); however, the underlying mechanisms have not yet been verified. Wu et al. focused on *miR-195*, the expression of which was significantly lower in placentas after the onset of PE than in uncomplicated placentas, and showed that this miRNA was highly expressed in the VT and EVT of the human early placenta by ISH [61]. In a bioinformatics analysis, they extracted activin receptor type-2B (*ActR2B*), transmembrane serine/threonine kinase receptor activin A belonging to the TGF-beta superfamily, as a candidate target of *miR-195*. Furthermore, a functional analysis using human EVT cell lines revealed that *miR-195* promoted EVT invasion by directly inhibiting ActR2B, indicating its involvement in early placentation.

### 2.2. lncRNAs

Few expression and functional analyses of lncRNAs in the human primary early placenta have been conducted to date. In our RNA sequencing analysis of EVT and CVT isolated from human primary early placentas, the expression of the lncRNAs *H19* and MALAT1 was predominant among all lncRNAs expressed [45]. In EVT, the expression levels of *H19* and *MALAT1* accounted for 70.5 and 16.7%, respectively, of total lncRNA expression levels, while in CVT, *H19* and *MALAT1* accounted for 22.0 and 32.8%, respectively. The expression of *H19* was approximately 3-fold higher in EVT than in CVT, suggesting that it plays a major role in the EVT invasion mechanism. Wang et al. showed that the lncRNA *MEG3* was highly expressed in the human early placenta by ISH [58]. They also demonstrated that the overexpression of *miR-210*, which is highly expressed in the early placenta as well as *MEG3*, in EVT cell lines resulted in the repression of *MEG3* and up-regulation of EMT-related genes, while the opposite was observed for the inhibition of *miR-210*. They concluded that *MEG3* is a negative regulator of EVT invasion; however, the underlying mechanisms have not yet been elucidated in detail. Collectively, these findings indicate that lncRNA *H19*, *MALAT1*, and *MEG3* are involved in early placentation and the development of PE.

## 3. Notable lncRNAs for Early Placentation

As described above, only a few studies have actually evaluated the lncRNAs that may play a role in early placentation using a gene expression or functional analysis of the primary human early placenta. We herein present three important lncRNAs, *H19*, *MALAT1*, and *MEG3*, as shown in Table 1 and Table 2.

### 3.1. H19

The lncRNA, *H19,* is located in a large imprinted *IGF2/H19* region with a length of approximately 2.0 kb on chromosome 11 [72]. *H19* and *IGF2* are epigenetically and reciprocally regulated, and *H19* is maternally expressed. *H19* is exclusively expressed in the embryo and placenta and markedly decreases after birth [73]. Previous studies reported the involvement of *H19* and *IGF2* in placental development. A dysregulation in imprinting of the *IGF2/H19* locus results in abnormal fetal/placental growth. The functional loss of the *H19* locus causes Beckwith–Wiedemann syndrome, a congenital overgrowth syndrome occasionally associated with embryonal tumors, including Wilms tumor, adrenal carcinoma, and neuroblastoma [74]. In contrast, the abolished function of *IGF2* is the cause of Silver–Russel syndrome, which is characterized by prenatal and postnatal growth restrictions with additional dysmorphic features, such as a triangular face with a small jaw and pointed chin, body asymmetry, and clinodactyly [74]. These findings suggest that *H19* acts as a “placental growth suppressor”. However, the underlying mechanisms remain unclear. On the other hand, *H19* is highly expressed not only in the placenta but also in various types of tumors. Previous studies showed that *H19* functions as an “oncogene” that accelerates cell proliferation, invasion, and metastasis in various manners, including miRNA sponging (e.g., for *let-7* and *miR-200*), the direct inactivation of p53, and guiding proteins (e.g., bringing EZH2, which participates in histone methylation, to the promotor region of *E-cad*, resulting in the inactivation of chromatin) [75,76,77,78]. Furthermore, *H19* has the unique function of “generating miRNA, *miR-675*” from exon 1 [73]. *miR-675* has been reported to be oncogenic or tumor suppressive, and, thus, its roles in tumorigenesis have been controversial. [79,80,81,82]. Tumorigenesis by *H19* appears to be inconsistent with its suppressive effects on fetal/placental growth. In addition to this contradiction, the biological roles and detailed functional mechanisms of *H19* have yet to be clarified. A previous study demonstrated that *H19* was up-regulated under hypoxic conditions [83]. As one of the mechanisms responsible, *H19* has a hypoxia-responsive element in its promotor region. In the course of normal placental development, sufficient EVT invasion is established in early pregnancy with the placenta being under hypoxic conditions [84], suggesting that *H19* plays a crucial role in EVT invasion.

Previous studies reported a relationship between *H19* and placental diseases. Xu et al. and Harati–Sadegh et al. performed a gene expression analysis between PE (*n* = 20 and 107, respectively) and control placentas (*n* = 20 and 113, respectively) using RT-PCR and showed that *H19* was significantly up-regulated in PE placentas [85,86]. In contrast, Gao et al. indicated that the expression level of *H19* was lower in EOPE placentas (*n* = 24) than in normal placentas (*n* = 24) [87]. They demonstrated that *miR-675* generated from *H19* inhibited *NOMO1*, which attenuated the proliferation of trophoblasts. We also reported that *miR-675-5p* generated from exon 1 of *H19* activated EVT invasion by promoting the expression of *MMP13* and *MMP14* through the inhibition of the transcriptional regulator *GATA2* using EVT cell lines [45]. Few studies have performed an expression analysis of *H19* using human early placentas. Zeng et al. showed that *H19* was highly expressed in trophoblast cells, particularly EVT, by ISH using human early placentas (6–12 weeks of gestation) and validated its expression by RT-PCR [62]. In addition, a microarray analysis using 12 abortion and 12 miscarriage samples showed that *H19* expression levels were significantly lower in miscarriage samples. These findings indicate that *H19* plays an important role in early embryogenesis and placentation; however, the effects of cell and tissue death need to be considered. As one of the underlying mechanisms, they demonstrated that *H19* might promote cell invasion by competing with *miR-106a-5p*, which directly inhibits VEGF, using functional analysis of EVT cell lines.

As described above, *H19* has a number of molecular functions. A more detailed understanding of the regulation of epigenetically expressed *H19* and the functional mechanisms of *H19* for placental development, including EVT invasion, will provide a breakthrough for the elucidation of the pathogenesis, prediction, and therapeutic options of not only PE but also other placental diseases, including fetal growth restriction and placenta accrete spectrum.

### 3.2. MALAT1: Metastasis-Associated Lung Adenocarcinoma Transcript 1

*MALAT1* is a long intergenic ncRNA encoded on chromosome 11q13.1 with a length of more than 8.0 kb and was initially shown to be associated with the metastasis of non-small cell lung cell cancer (NSCLC) [88]. *MALAT1* is highly conserved among mammals and is ubiquitously expressed in almost all organs. The main function of *MALAT1* is the regulation of gene expression. *MALAT1* is mainly localized in the nucleus and plays a role in “post-transcriptional regulation” by being recruited to nuclear paraspeckles and regulating the alternative splicing of a number of pre-mRNAs through the modulation of serine-/arginine-rich splicing factor phosphorylation [89,90]. *MALAT1* also participates in “transcriptional regulation” by acting as a guide bringing PR2 (a member of the polycomb repressive complex), which modulates the methylation of the histone, to the promotor region of target genes [91]. These functions suggest that *MALAT1* acts as either an activator or suppressor of target genes. However, the detailed functions of *MALAT1* with every tissue and cell type have yet to be clarified.

Due to its ubiquitous and enriched expression, *MALAT1* has been examined by many researchers in order to elucidate the pathogenesis of various diseases, including cancers, cardiovascular disease, and diabetes [92,93,94]. Oncology reports for *MALAT1* have been increasing. *MALAT1* expression was found to fluctuate not only in NSCLC but also in various cancers. *MALAT1* has been identified as a negative prognostic marker in many types of cancers (e.g., gastric cancer, colorectal cancer, renal cell carcinoma, and hepatocellular carcinoma after liver transplantation) [95,96,97,98]. Some researchers indicated the oncogenic functional mechanisms of *MALAT1*. Hirata et al. reported that *MALAT1* promoted EMT in renal cell carcinoma cell lines by accelerating the methylation of the promotor region of *E-cad* by recruiting EZH2 (polycomb protein) [97]. In bladder cancer, Fan et al. demonstrated that *MALAT1* promoted EMT by inducing the expression of *TGFB* using bladder cancer cell lines and a mouse model [99]. In contrast, *MALAT1* was shown to function as a tumor suppressor. Vassallo et al. demonstrated that *MALAT1* inhibited cell migration by attenuating the non-canonical WNT pathway in vitro using human glioblastoma cell lines and in vivo using a mouse glioblastoma model [100]. Xu et al. revealed that the expression of *MALAT1* was lower in human breast cancer tissues than in non-tumor tissues, and *MALAT1* suppressed EMT via the PI3K-AKT pathway using breast cancer cell lines [101]. These findings suggest that the effects of *MALAT1* for tumorigenesis are dependent on tissue and cell types.

Limited information is currently available on the relationship between *MALAT1* and placentation, including the pathogenesis of PE. Chen et al. and Li et al. showed that the expression of *MALAT1* was down-regulated in PE placental cells and mesenchymal stem cells (MSCs) from the PE umbilical cord, respectively [102,103]. They demonstrated that *MALAT1* promoted cell proliferation, migration, invasion, and the cell cycle using trophoblast cell lines and MSCs. Collectively, these findings imply that *MALAT1* plays a crucial role in placental development as an accelerator of EVT invasion; however, a *MALAT1* expression analysis using human early primary placentas has yet to be conducted.

### 3.3. MEG3: Maternally Expressed Gene 3

*MEG3* is a maternally expressed imprinted gene in the *DLK1-MEG3* locus on human chromosome 14q32.3 that encodes lncRNA. *MEG3* transcribes multiple variants, and the length of the main isoform is approximately 13.0 kb. *MEG3* is highly expressed in the placenta and adrenal glands but is ubiquitously expressed in normal human tissues. Many studies have demonstrated that *MEG3* functions as a tumor suppressor. Zhang et al. showed that *MEG3* inhibited cell growth, invasion, and angiogenesis in breast cancer cells by attenuating the PI3K/AKT pathway [104]. Furthermore, *MEG3* activated *p53* transcription by inhibiting *MDM*, which mediated the degradation of p53 in human colorectal cancer and osteosarcoma cell line models [105]. In meningioma, *MEG3* accelerates tumor suppressor *RB1* directly or indirectly by activating *CDKN4A*, resulting in tumor cell growth arrest [106]. These findings showed that *MEG3* acts as a tumor suppressor by interacting with a number of well-characterized tumorigenesis-related genes, such as *VEGFA*, *TGFB1*, *p53*, *MDM2*, *GDF15*, and *RB1*. *MEG3* also serves as a tumor suppressor by “miRNA sponging”. In breast cancer cells, *MEG3* has been shown to modulate the inhibition of *E-Cad* by sponging *miR-421*, resulting in the suppression of EMT [107]. Xu et al. and Dan et al. demonstrated that *MEG3* inhibited tumorigenesis, including cell proliferation, invasion, and metastasis, via the sponging of *miR-21*, known as an oncogenic miRNA, in gastric cancer [108,109].

Many researchers have investigated the role of *MEG3* in placental development. As described above, Wang et al. reported that the mechanism by which decreased *miR-210* expression during the differentiation of CVT to EVT resulted in accelerated EVT invasion was due to the inhibitory effects of *miR-210* on *MEG3* [58]. Zhang et al. showed that *MEG3* expression was lower in PE placentas (*n* = 30: mean 35.9 w) than in normotensive placentas (*n* = 30: mean 38.1 w) [110]. They also demonstrated that MEG3 reduced cell apoptosis and promoted cell migration via NFκB signaling in the trophoblast cell lines, HTR-8/SVneo and JEG3. Although these functions of *MEG3* appeared to be oncogenic, they speculated that inadequate trophoblast functions due to a decrease in *MEG3* were associated with failed uterine spiral artery remodeling, which contributed to the development of PE. Liu et al. focused on the relationship between *MEG3* and the functions of decidual natural killer (dNK) cells [111]. To examine the mechanisms underlying uterine spiral artery remodeling, they cocultured vascular smooth muscle cells (VSMCs) and dNK cells with IFNγ and investigated changes in the proliferation, migration, and apoptotic abilities of VSMCs. The dNK/IFNγ treatment promoted the migration of VSMCs via the up-regulation of *MEG3*. They also suggested that *MEG3* may accelerate trophoblast invasion. The inconsistent roles of trophoblasts and tumor cells in cell invasion are interesting, and, thus, functional analyses of *MEG3* are needed. Furthermore, the imprinting mechanisms of *MEG3* should be extensively elucidated in the future.

## 4. Possible Circulating ncRNA Biomarkers for PE in the First Trimester

Pregnant women with PE may show the abnormal expression of ncRNAs involved in EVT invasion in the placenta in the early stage of pregnancy. However, this is very difficult to demonstrate for the following reasons: (1) CVS is rarely performed, except to test for fetal congenital diseases, and (2) it is challenging to follow the final pregnancy outcomes of cases that underwent CVS in early pregnancy. It is not practical to perform CVS for the purpose of predicting PE. Therefore, difficulties are currently associated with the clinical application of ncRNA expression levels in chorionic villi or placenta in early pregnancy as a biomarker for PE. On the other hand, ncRNAs expressed in the placenta are detectable in the maternal circulation, some of which become difficult to detect after delivery [52,112]. Therefore, high expression levels of ncRNAs in the placenta, particularly chorionic villi and EVT cells, may affect expression levels in maternal blood throughout pregnancy. Smith et al. performed a miRNA sequencing analysis of placentas and plasma from non-complicated pregnant women at 6–23 weeks of gestation, and the findings obtained confirmed that placental top-ranked highly expressed miRNAs were also highly expressed in plasma [47]. In addition, the expression of placenta-specific miRNAs, such as C19MC and C14MC miRNAs, in the placenta was similar to that in the plasma during gestation. Therefore, if failed EVT invasion (the etiology of PE) is observed in early pregnancy, changes in the expression of genes, including ncRNAs, involved in EVT invasion may also be detected in the early maternal circulation. A marked difference in the expression levels of ncRNAs in blood between pregnant women who will develop PE and those who will not at the early stage of pregnancy may be clinically applied as a predictive biomarker for PE in early pregnancy. We herein summarize previous findings on ncRNAs associated with the later development of PE or with the potential to be used as biomarkers in analyses of blood samples from early pregnant women (Table 3) [113,114,115,116,117,118,119,120,121,122,123,124,125,126].

### 4.1. miRNAs

#### 4.1.1. Placenta-Specific miRNA Clusters

Hromadnikova et al. focused on C19MC miRNAs, a placenta-specific miRNA cluster, in maternal blood samples in early pregnancy [113,114]. They previously reported that the combined score of 3 C19MC miRNAs: *miR-516-5p*, *miR-520h*, and *miR-518b*, in early maternal plasma was associated with the development of later gestational hypertension [127]. They then retrospectively quantified 6 C19MC miRNAs in the plasma of 21 pregnant women who later developed PE and 58 normal pregnant women collected at 10–13 weeks of gestation by RT-PCR and showed that *miR-517-5p*, *miR-518b*, and *miR-520h* were significantly up-regulated in the plasma of pregnant women who later developed PE. *miR-517-5p* was the most accurate marker in terms of predicting the onset of PE, with an area under the curve (AUC) of 0.70 (*p* = 0.045), sensitivity of 43%, and specificity of 86.2% [113]. They also analyzed exosome-derived C19MC miRNA in maternal plasma. Exosomes, a type of extracellular vesicle, are 30–150-nm vesicular structures that are released from a number of cells, including trophoblasts. Exosomes contain cell-derived proteins, mRNAs, and ncRNAs and have been shown to play an important role in cell-to-cell communication and signaling [128,129]. Kambe et al. demonstrated that *miR-517a-3p* (C19MC miRNA) in exosomes repressed target genes through cell-to-cell communication by an in vitro model using trophoblast cell lines and peripheral blood NK cells [130]. Hromadnikova et al. retrospectively collected exosomes from maternal plasma at 10–13 weeks from 43 pregnant women who later developed PE and 102 normal pregnant women using an exosome isolation kit (not the ultracentrifugation method), as previously reported, and evaluated C19MC miRNA expression by RT-PCR [114]. In contrast to their previous findings, the expression levels of C19MC miRNAs in exosomes obtained from maternal plasma were slightly lower in pregnant women who later developed PE than in normal pregnant women, with *miR-520a-5p* and *miR-525-5p* being significantly lower. In terms of the prediction of PE, *miR-520a-5p*, *miR-525-5p*, and *miR-517-5p* alone and the combination of these 3 miRNAs in plasma-derived exosomes showed a sensitivity of more than 50–60% and a specificity of more than 70–80%, which was more accurate than the analysis of plasma itself (not exosomal miRNAs) [114]. The reason for the discrepancy in the direction of gene expression fluctuations between plasma and plasma exosomes remains unclear and, thus, needs further study. Jiang et al. evaluated the expression level of *miR-520g*, a C19MC miRNA, in the serum of 19 pregnant women who later developed PE and the same number of serum samples from normal pregnant women and showed that *miR-520g* expression levels were significantly higher in the later developed PE group in the first trimester, in contrast to the second and third trimesters [115]. They also demonstrated using functional analysis of EVT cell lines that *miR-520g* inhibited cell invasion by suppressing MMP2. These findings suggest that the high expression of *miR-520g* in the placenta during early pregnancy is involved in failed EVT invasion, leading to the development of PE and that *miR-520g* has potential as a serum biomarker for the prediction of PE in the first trimester.

**Table 3 cells-11-02428-t003:** Possible maternal circulating non-coding RNA biomarkers for preeclampsia in the 1st trimester obtained from case control studies.

No	Author	Country	Study Outcome	Race	Sampling Time (GW)	Sample	Case (n)/Control (n)	Predictive ncRNAs	Type of Predictive ncRNA	Quantitative Method	Expression Level	Accuracy					Reference
												Sensitivity (%)	Specificity (%)	PPV (%)	NPV (%)	AUC	
1	Hromadnikova (2017)	Czech	PE	Caucasian	10–13 w	Plasma	PE (21)/Normal (58)	*miR-517-5p* (C19MC)	miRNA	RT-PCR	Up in PE	42.9	86.2	52.9	80.6	0.70	[113]
								*miR-518b* (C19MC)			Up in PE	52.4	63.8	34.4	78.7	0.55	
								*miR-520h* (C19MC)			Up in PE	14.3	96.6	60.0	75.7	0.45	
2	Hromadnikova (2019)	Czech	PE	Caucasian	10–13 w	Exosomes from plasma	PE (43)/Normal (102)	*miR-520a-5p* (C19MC)	miRNA	RT-PCR	Down in PE	60.5	84.0	26.0	42.0	0.69	[114]
								*miR-525-5p* (C19MC)			Down in PE	51.2	84.0	22.0	42.0	0.69	
								*miR-517-5p* (C19MC)			Down in PE	60.4	70.0	26.0	35.0	0.63	
								3 miRNAs			Down in PE	65.1	78.0	28.0	39.0	0.71	
3	Jiang L (2017)	China	PE	NA	8–10 w	Serum	PE (19)/Normal (19)	*miR-520g*	miRNA	RT-PCR	Up in PE	NA	NA	NA	NA	NA	[115]
4	Ura (2014)	Italy	PE	Multiracial (Caucasian: > 90%)	12–14 w	Serum	PE (24)/Normal (24)	*miR-1233*	miRNA	RT-PCR	Up in PE (5.4x)	NA	NA	NA	NA	NA	[116]
								*miR-520a*			Up in PE (3.2x)	NA	NA	NA	NA	NA	
								*miR-210*			Up in PE (3.3x)	NA	NA	NA	NA	NA	
								*miR-144*			Down in PE (0.4x)	NA	NA	NA	NA	NA	
5	Licini (2021)	Italy	PE	NA	12 w	Plasma	PE (13)/Normal (18)	*miR-125b*	miRNA	RT-PCR	Down in PE					0.85	[117]
6	Martinez-Fierro (2021)	Mexico	PE	NA	12 w	Serum	PE (6)/Normal (6)	*miR-628-3p*	miRNA	Taqman low density array	Up in PE (7.7x)	NA	NA	NA	NA	NA	[118]
7	Martinez-Fierro (2019)	Mexico	PE	NA	12 w	Serum	PE (16)/Normal (18)	*miR-628-3p*	miRNA	RT-PCR	Up in PE (7.7x)	NA	NA	NA	NA	NA	[119]
8	Chen (2021)	China	PE	NA	11–13 w	Whole blood	PE (24)/Normal (30)	*has_circ_0025992*	circRNA	RT-PCR	Up in PE	54.2	93.3	86.7	71.8	0.80	[120]
9	Timofeeva (2018)	Russia	EOPE	NA	11–13 w	Plasma	EOPE (6)/Normal (10)	*miR-423-5p*	miRNA	RT-PCR	Down in PE	NA	NA	NA	NA	NA	[121]
								*miR-532-5p*			Down in PE	NA	NA	NA	NA	NA	
10	Winger (2015)	USA	PE	NA	preconception-9 w	PBMC	PE (12)/Normal (19)	Score using 7 miRNAs (*miR-1*, *miR-133b*, *miR-199a-5p*, *miR-1267*, *miR-1229*, *miR-148a-3p*, *miR-223*)	miRNA	RT-PCR (Scoring)	NA	83.3	89.5	83.0	89.0	0.90	[122]
			EOPE				EOPE (5)/Normal (19)				NA	80.0	89.5	67.0	94.0	0.86	
11	Winger (2018)	UK	PE	Multiracial	11–13 w	Plasma	PE (4)/Normal (19)	*miR-1267*	miRNA	RT-PCR	NA	75	95	NA	NA	0.88	[123]
							PE (2)/Normal (7)	*miR-148a*			NA	100	86	NA	NA	0.93	
							PE (2)/Normal (4)	*miR-196a*			NA	100	100	NA	NA	1.00	
							PE (3)/Normal (8)	*miR-33a*			NA	100	100	NA	NA	1.00	
							PE (3)/Normal (9)	*miR-575*			NA	100	78	NA	NA	0.93	
							PE (7)/Normal (9)	*miR-582*			NA	100	100	NA	NA	1.00	
							PE (3)/Normal (14)	*miR-210*			NA	67	100	NA	NA	0.86	
							PE (4)/Normal (20)	*miR-16*			NA	100	55	NA	NA	0.78	
							PE (8)/Normal (40)	Score using 8 miRNAs		(Scoring)	NA	75	90	NA	NA	0.91	
12	Yoffe (2018)	Israel	EOPE	Multiracial	11–13 w	Plasma	EOPE (35)/Normal (40)	Score using 25 ncRNAs (12 miRNAs, 4 lncRNAs, 1 rRNA, 7 mitochondrial tRNA, 1 processed transcript)	miRNA, lncRNA, rRNA, mitochondrial tRNA, processed transcript	Small RNA sequencing	NA	72.0	80.0	NA	NA	0.86	[124]
13	Dai (2021)	China	PE	NA	<20 w *	Serum	PE (97)/Normal (97)	*NR_002187*	lncRNA (pseudogene)	RT-PCR	Up in PE	NA	NA	NA	NA	0.66	[125]
								*ENST00000415029*	lncRNA		Up in PE	NA	NA	NA	NA	0.69	
								*ENST00000398554*	lncRNA		Up in PE	NA	NA	NA	NA	0.65	
								*ENST00000586560*	lncRNA		Up in PE	NA	NA	NA	NA	0.63	
								*TCONS_00008014*	lncRNA		Up in PE	NA	NA	NA	NA	0.64	
								*ENST00000546789*	lncRNA		Up in PE	NA	NA	NA	NA	0.68	
								*ENST00000610270*	lncRNA		Up in PE	NA	NA	NA	NA	0.66	
								*ENST00000527727*	lncRNA		Up in PE	NA	NA	NA	NA	0.64	
14	Tarca (2021)	USA	EOPE	Multiracial (African American > 90%)	11–17 w	Whole blood	EOPE (13)/Normal (49)	H19	lncRNA (lincRNA)	Z score obtained from microarrays (EOPE vs. Normal)	Up in PE	NA	NA	NA	NA	NA	[126]

C19MC: chromosome 19 microRNA cluster, EOPE: early onset preeclampsia, lncRNA: long non-coding RNA, miRNA: microRNA, NA: not available, ncRNA: non-coding RNA, PBMC: peripheral blood mononuclear cell, PE: preeclampsia, rRNA: ribosomal RNA, RT-PCR: real time-polymerase chain reaction, tRNA: transfer RNA * including samples after the 1st trimester.

#### 4.1.2. Hypoxia-Responsive miRNA: miR-210

As described in the previous paragraph, the hypoxia-inducible miRNA, *miR-210,* is highly expressed in the early placenta, and its expression decreases during the differentiation of VT into EVT [58]. The *miR-210* expression status appears to be associated with the invasive ability of EVT and early placentation. Ura et al. showed that *miR-210* was significantly up-regulated in the serum of early pregnant women who later developed PE [116]. Winger et al. incorporated *miR-210* as one component of a PE prediction model using 8 miRNAs in the plasma of early pregnant women (described in the following paragraph) [123]. Although limited information is currently available, *miR-210* has potential as a circulating predictive biomarker of PE in early pregnancy.

#### 4.1.3. Circulating Predictive miRNAs of PE Based on Gene Expression Analyses Using Human Samples

Other researchers extracted circulating PE biomarker candidates based on gene expression analyses using placenta and blood samples from pregnant women who had already developed PE. To identify ncRNAs predictive of PE, the majority of studies analyzed the outcome as the development of PE, not clinically important EOPE or POPE. Licini et al. focused on miRNAs, which were previously shown to be involved in angiogenesis, cell migration, and vascular remodeling in oncology studies, and retrospectively compared the expression levels of *miR-125b* in the plasma of pregnant women who later developed PE and those who did not at 12 weeks of gestation by RT-PCR [117]. *miR-125b* was significantly up-regulated in the plasma of women who later developed PE. They also indicated the potential of the log value of plasma *miR-125b* expression levels in early pregnancy combined with maternal age and BMI as a predictive model for the onset of PE (AUC 0.85). Furthermore, in a functional analysis using primary trophoblast cells and trophoblast cell lines, they demonstrated that miR-125b was highly expressed in SCT and suppressed the expression of *TACSTD2* (tumor-associated calcium signal transducer (2)), which is responsible for cell growth, thereby providing a model for its involvement in the development of PE [117]. Martinez–Fierro et al. focused on *miR-628-3p*, which regulates EVT invasion by directly inhibiting *Runx2*, which is highly expressed in EVT and promotes the expression of matrix metalloproteinases such as *MMP2* and *MMP9*. They then retrospectively compared the expression levels of *miR-628-3p* at 12 weeks of gestation in 16 pregnant women who later developed PE and 16 pregnant women who did not by microarrays and RT-PCR [118,119]. The expression of *miR-628-3p* was 7.7-fold higher in pregnant women who later developed PE than in those without PE, indicating the potential of *miR-628-3p* as a biomarker for predicting the onset of PE. Chen et al. performed a microarray analysis on endothelial cells obtained from the umbilical cords of pregnant women who developed PE in the third trimester (*n* = 4) and those who did not (*n* = 4) immediately after delivery and identified genes with significantly altered expression levels [*p* < 0.05, FC ≥ 2.0: 33 mRNAs, 272 circular RNAs (circRNAs), and 207 lncRNAs]. They conducted a bioinformatics analysis of these genes and extracted 2 mRNAs, 5 circRNAs, and 3 lncRNAs as candidate biomarkers for the prediction of PE, which are the core of the network responsible for biological functions, such as cell growth, endothelial cell migration, and metabolism [120]. Although the number of validated samples was small, a retrospective analysis of the expression levels of these genes in the plasma of early pregnant women (11–13 weeks of gestation) showed that *hsa-circ-0025992* was significantly predictive of PE with a sensitivity of 54% and specificity of 93%. Timofeeva et al. performed a microarray analysis of placentas from 16 normal pregnant women, 16 EOPE pregnant women, and 12 late-onset preeclampsia (onset at >34 w: LOPE) pregnant women and extracted 7 miRNAs as candidate PE biomarkers with significantly altered expression levels (*p* < 0.05, FC ≥ 2.0) in both EOPE and LOPE from those in normal pregnancy [121]. In addition, expression analysis of these candidate genes in the plasma at each trimester (first, second, and third trimesters) in 6 pregnant women who later developed EOPE and 10 normal pregnant women revealed the significantly lower expression of *miR-423-5p* and *miR-532-5p* at 11–13 weeks of gestation, indicating their association with EOPE [121]. This study also had a small number of validated pregnant women; nevertheless, the findings obtained are of value because they showed the relationship between the expression of miRNAs (*miR-423-5p* and *miR-532-5p*) in the circulation in early pregnancy and clinically important EOPE. Winger et al. focused on miRNAs with significantly altered expression levels by applying a microarray analysis to the peripheral blood of pregnant women with PE or miscarriage and those of normal early pregnant women and previously reported miRNAs that regulate inflammatory signals. They also created a score model using seven miRNAs (*miR-1*, *miR-133b*, *miR-199a-5p*, *miR-1267*, *miR-1229*, *miR-148a-3p*, and *miR-223*) [122]. This score model was validated using peripheral blood samples from 12 pregnant women who later developed PE between 14 weeks before implantation to 9 weeks of gestation (5 of whom developed EOPE) and 19 normal pregnant women. This model predicted PE and EOPE with sensitivities of 83.3 and 80%, respectively, and the same specificity of 89.5% (AUC 0.90 and 0.86, respectively). Furthermore, they focused their analysis on plasma samples at 11–13 weeks of gestation and showed that 8 miRNAs (*miR-1267*, *miR-148a*, *miR-196a*, *miR-33a, miR-575*, *miR-582*, *miR-210*, and *miR-16*) were associated with the development of PE [123]. Ura et al. performed a microarray analysis of sera from 24 pregnant women who later developed severe PE in the third trimester and 24 pregnant women who did not at 12–14 weeks of gestation and identified 19 miRNAs with significantly altered expression levels (*p* < 0.05, FC ≥ 2.0) as candidate biomarkers for the prediction of PE [116]. RT-PCR analysis of these candidate miRNAs in early maternal sera showed that *miR-1233* was 5.4-fold higher, *miR-520a* (C19MC miRNA) was 3.2-fold higher, and *miR-210* (hypoxia-inducible miRNA) was 3.3-fold higher in pregnant women who later developed severe PE in the third trimester than in those who did not. On the other hand, the expression levels of *miR-144* were 2.5-fold higher in pregnant women without PE. These findings suggested relationships between these miRNAs and the development of PE retrospectively, although a predictive ability was not demonstrated. Yoffe et al. performed an RNA sequencing analysis of plasma samples at 11–13 weeks of gestation of 35 pregnant women who later developed EOPE and 40 normal pregnant women at 11–13 weeks of gestation and identified 25 ncRNAs with significantly altered expression levels, including 12 miRNAs (adjusted *p* < 0.05) as candidate biomarkers for the prediction of EOPE [124]. These ncRNAs showed similar changes in expression levels when evaluated by RT-PCR. They revealed that EOPE might be predicted in the first trimester with a sensitivity of 72% and specificity of 80% (AUC 0.86) by the combined score of these ncRNAs (expression levels of 12 miRNAs, 4 lncRNAs, 1 rRNA, 7 mitochondrial transfer RNA, and 1 processed transcript).

### 4.2. lncRNAs

Few studies have evaluated the predictive ability of lncRNA for PE using blood samples collected from early pregnant women. Dai et al. reported the PE predictive ability of lncRNAs using serum samples obtained at less than 20 weeks of gestation; however, samples outside the first trimester were also included [125]. They performed a microarray analysis of the sera of 5 pregnant women who later developed PE and 5 pregnant women who did not, and lncRNAs (up-regulated: 417 lncRNAs, down-regulated: 167 lncRNAs) and miRNAs (up-regulated 510 miRNAs, down-regulated 226 miRNAs) with significantly altered expression levels were subjected to bioinformatics analysis. The top 8 lncRNAs (*NR_002187*, *ENST00000415029*, *ENST00000398554*, *ENST00000586560*, *TCONS_00008014*, *ENST00000546789*, *ENST00000610270*, and *ENST0000527727*) were selected as candidate biomarkers for the prediction of PE from the experimentally supported node genes of the disease (PE)-based network. The expression levels of these lncRNAs in early pregnancy (less than 20 weeks of gestation) were elevated in pregnant women who later developed PE, indicating a relationship with the development of PE. However, the functions of these lncRNAs were not elucidated. Tarca et al. investigated 16 genes, including lncRNA *H19*, which were significantly up-regulated in the peripheral blood of EOPE patients at the time of onset, as potential biomarkers for the prediction of EOPE, and retrospectively analyzed gene expression in the sera of 13 pregnant women who later developed EOPE and 49 normal pregnant women at early, mid-, and full-term pregnancy [126]. The findings obtained showed that several predictive scores from multiple combinations of the log values of 4 genes (*H19*, *FN1*, *TUBB6*, and *FPR3*: calculated from the difference between the values of cases and those of non-pregnant women in the microarray analysis) out of 16 possible biomarkers were predictive of the onset of EOPE at 22–28 weeks of gestation. However, those scores were not significantly predictive of EOPE in early pregnancy. On the other hand, the expression of lncRNA *H19* was significantly higher in pregnant women who later developed PE than in those who did not at 11–17 weeks of gestation, suggesting a relationship between *H19* and the pathogenesis and onset of EOPE [126]. Therefore, as described in the previous section, *H19* is expected to be validated as a PE biomarker in early pregnancy.

It currently remains unclear whether plasma or serum samples are more useful for evaluating circulating ncRNAs. In miRNA-sequencing analyses, the top-ranked maternal circulating miRNAs do not significantly differ between plasma and serum; however, the expression levels of some miRNAs appear to differ markedly. Although plasma may be more suitable for a circulating ncRNA expression analysis because of enriched mapping reads, further verification is needed [131].

## 5. Future Perspectives

In this review, we focused on the findings of expression analyses of ncRNAs in the placenta and maternal blood in early pregnancy. Although the ncRNAs described herein have potential as first trimester PE predictors, limited information is currently available. A number of issues need to be resolved for the establishment of ncRNA-based models that extract high-risk groups of PE, particularly POPE. As described above, in consideration of invasion for the mother and fetus (placenta), the most clinically feasible approach appears to be the development of a predictive model using circulating ncRNA in early pregnancy alone or in combination with multiple factors, including circulating ncRNAs (Figure 2). These ncRNAs were (1) highly expressed in the early placenta and/or (2) significantly altered expression levels in the peripheral blood of pregnant women who later developed PE in early pregnancy. It is important to note that none of these have yet fully demonstrated the predictive ability for the high-risk group of PE consistently in multiple reports with reproducibility. Although these ncRNAs have the potential to predict PE, at present, they are still limited to ncRNAs that are associated with the development of PE. ncRNAs that show stronger association and predictive ability for the development of PE are desired and methods for their extraction and validation must be devised. In addition to C19MC/C14MC miRNAs, *miR-210*, and lncRNA *H19* described herein, many ncRNAs are highly expressed in the early placenta and villi and may be involved in EVT invasion. As shown in the studies presented in this review, the identification of more novel candidate biomarkers is needed using bioinformatics analysis. An analysis of ncRNA expression in trophoblasts, EVT, and decidual cells in early pregnancy by single-cell sequencing may provide important insights. More detailed gene expression analyses of early placentas and evaluations, including final pregnancy outcomes, will increase the possibility of extracting circulating biomarker candidates for the prediction of PE; however, the use of a high number of cases is ethically challenging. In parallel, further studies on the functions of these PE predictive ncRNAs are needed. Limited information is currently available on the expression and function of lncRNAs in the early placenta, and various validation reports are expected. In addition, prospective cohort studies with more blood samples from early pregnant women are needed for the development of candidate models for ncRNA-based PE predictions. In previous analyses of ncRNA expression in the blood of early pregnant women described in this review, many set the onset of all types of PE as the study outcome. The clinically important outcome is POPE, particularly EOPE, which is strongly influenced by the prematurity of the fetus/infant due to an early delivery. Therefore, analyses with the outcome of POPE or EOPE are desirable. Prospective cohorts require the accumulation of a large amount of case data since the incidence of POPE is approximately 1–2%. In POPE, EVT invasion, which is one of the most important contributing factors to pathogenesis, is predominantly active between 9–13 weeks. Considering the period of EVT invasion in early placentation, it may be possible to extract the ncRNAs more strongly associated with PE development by the verification using maternal blood samples by 9–13 weeks of gestation. Another challenge to achieving the prediction of PE in high-risk pregnant women using circulating ncRNAs is the issue of cost. RT-PCR and digital PCR used for quantification of maternal circulating ncRNAs are more expensive than those of serum PlGF used in the FMF model. Circulating ncRNA should be evaluated in all early pregnant women for the screening of the PE high-risk group. Therefore, in order to achieve the ncRNA-based PE high-risk prediction model, not only the accuracy of the prediction ability but also the cost-effectiveness compared to the conventional FMF model must be verified. If a useful ncRNA PE prediction model is established, it will be combined with the existing POPE/EOPE high-risk group extraction model (e.g., FMF model) to further improve its accuracy, leading to the prevention of POPE and EOPE by LDA.

## Figures and Tables

**Figure 1 cells-11-02428-f001:**
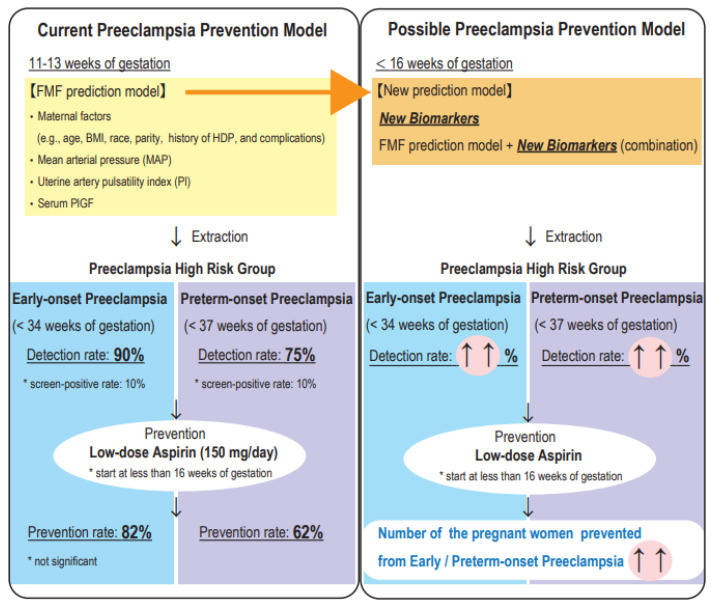
Current and possible prevention models for preeclampsia. The current model, validated by the ASPRE trial, uses a combination of maternal characteristics and mean maternal arterial pressure, the uterine artery pulsatility index, and maternal serum PlGF at 11–13 weeks of gestation to identify pregnant women at high risk of preterm onset preeclampsia (POPE). A low oral dose aspirin (150 mg/day) for these women before 16 weeks of gestation reduced POPE by 62% (significant) and early onset preeclampsia by 82% (not significant). In this model, the detection rate of POPE high-risk pregnant women was 75%, and increasing this sensitivity is our challenge. New biomarkers that extract POPE or EOPE high-risk pregnant women with high sensitivity before 16 weeks of gestation are desired.

**Figure 2 cells-11-02428-f002:**
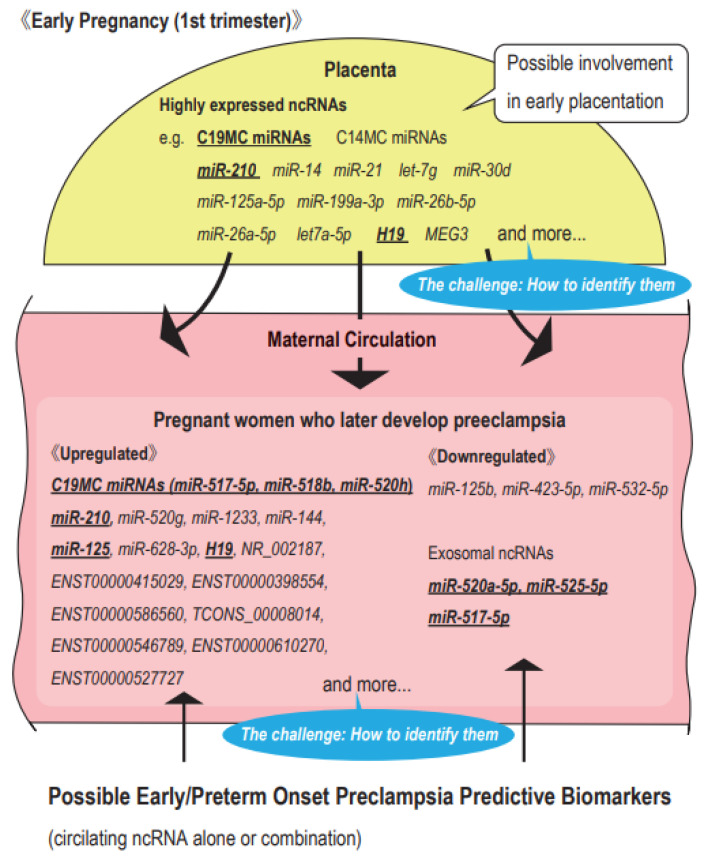
Possible predictive biomarkers for early/preterm onset preeclampsia. Highly expressed non-coding RNAs in the placenta or extravillous trophoblasts in early pregnancy may be involved in early placentation and in the pathogenesis of early-onset and preterm onset preeclampsia (EOPE and POPE). Some of these ncRNAs are also detected in the maternal circulation. ncRNAs that show significant differences in their expression levels and patterns in early pregnancy between normal pregnant women and those who later develop EOPE or POPE have potential as predictive biomarkers for EOPE or POPE in early pregnancy. Under bar: ncRNAs that are highly expressed in the early placenta and show significantly altered expression levels in the peripheral blood of early pregnant women who later develop PE.

**Table 1 cells-11-02428-t001:** Highly expressed non-coding RNAs in the human early placenta (Profiles of non-coding RNAs in 1st trimester placentas).

Author	Type of ncRNA	Study Method	Features	Sampling Time	Sample Size	Reference
Morales-Prieto (2012)	miRNA	Microarray analysis (1st vs. 3rd trimester placentas) ↓ Validation by RT-PCR	・C19MC miRNAs significantly increased from 1st to 3rd trimester trophoblasts. ・C14MC miRNA significantly decreased from 1st to 3rd trimester trophoblasts. ・*miR-14*, *miR-21*, and *let-7g* were highly expressed throughout pregnancy. ・In the clustering analysis, C14MC miRNAs were close to JEG3, a choriocarcinoma cell line, while C19MC miRNAs were close to HTR-8/SVneo, an EVT cell line.	1st trimester	3	[51]
				3rd trimester	3	
Luo (2009)	miRNA	Small RNA library sequencing	・30–40% of the miRNAs highly expressed in 1st trimester placentas were C19MC miRNAs.	1st trimester (7–11 w)	placenta: 6, plasma: 3	[44]
				3rd trimester (36–38 w)	placenta: 6, plasma: 3	
		ISH	・C19MC miRNAs were more highly expressed in SCT than in CVT.	1st trimester		
		RT-PCR analysis of C19MC miRNAs in adult organs	・*miR-517a*, *miR-517b*, *miR-518b*, *miR-519A*, and *miR-512-3p* were expressed exclusively in the placenta among adult organs.	NA		
		RT-PCR comparative analysis of C19MC miRNAs in maternal plasma (before vs. after delivery)	・The expression levels of C19MC miRNAs (*miR-517a* and *miR-518b*) in maternal plasma markedly decreased after delivery.	1 day before delivery 3 days after delivery	6 6	
Miura (2010)	miRNA	Microarray analysis (1st vs. 3rd trimester placentas/plasma)	・Among 82 placenta-predominant miRNAs (>100-fold higher in the placenta than in plasma), C19MC miRNAs and C14MC miRNAs accounted for 53.7 and 15.9%, respectively.	1st trimester (12–13 w) 3rd trimester (38–39 w)	placenta: 2, plasma: 2 placenta: 2, plasma: 2	[52]
		RT-PCR comparative analysis using plasma samples (1st vs. 3rd trimester vs. post delivery)	・Among placenta-predominant miRNAs, 24 miRNAs significantly decreased after delivery, particularly *miR-515-3p*, *miR-517a*, *miR-517c*, *miR-518b*, *miR-526b* (C19MC), and *miR-323-3p* (C14MC).	1st trimester (12–13 w) 3rd trimester (38–39 w) Day 1 after delivery	10 10 10	
Ogoyama (2021)	miRNA	RNA sequencing for EVT and CVT isolated from human early placentas	・The expression of C14MC miRNAs was significantly lower in EVT than in CVT.	7 w	3	[45]
	lncRNA		・The expression level of lncRNA *H19* accounted for 70% of all lncRNAs in EVT.			
Gonzalez (2021)	miRNA	miRNA sequencing (1st vs. 3rd trimester placentas)	・C19MC miRNAs significantly increased from 1st to 3rd trimester trophoblasts. ・C14MC miRNA significantly decreased from 1st to 3rd trimester trophoblasts.	1st trimester 3rd trimester	113 47	[46]
Smith (2021)	miRNA	miRNA sequencing using placentas and plasma from 6–23 w of gestation. ↓ Expression comparison (6–10 w vs. 11–23 w)	・The top 10 miRNAs with the highest expression levels in the placenta were *miR-30d-5p*, *miR-125a-5p*, *miR-517a-3p**, *miR-199a-3p*, *miR-26b-5p*, *miR-26a-5p*, *let-7a-5p*, *miR-21-5p*, *miR-126-3p*, and *miR-516b-5p** (C19MC*). ・Thirty-four out of 48 C19MC miRNAs (75%) were downregulated in 11–23 w of gestation. ・Fifty-six out of 77 C14MC miRNAs (73%) were up-regulated in 11–23 w of gestation. ・The top highest miRNAs in the placenta were also highly expressed in plasma.	6–10 w 11–23 w	total 86	[47]

CVT: chorionic villous trophoblast, C14MC: chromosome 14 microRNA cluster, C19MC: chromosome 19 microRNA cluster, CVT: chorionic villous trophoblast, EVT: extravillous trophoblast, ISH: in situ hybridization, lncRNA: long non-coding RNA, miRNA: microRNA, RT-PCR: real time-polymerase chain reaction, SCT: syncytiotrophoblast.

**Table 2 cells-11-02428-t002:** Highly expressed non-coding RNAs in the human early placenta (Validation of the expression levels of non-coding RNAs using 1st trimester placentas).

No	Author	ncRNA	Type of ncRNA	Study Method	Features	Sampling Time	Sample Size	Reference
1	Wang (2012)	*miR-517b*	miRNA (C19MC)	ISH and RT-PCR	・*miR-517b* and *miR-519a* were located in the trophoblast layer.	6–9 w	ISH: 1 in each GW	[53]
		*miR-519a*					RT-PCR: 10 in each GW	
2	Mong (2020)	*miR-517a miR-517c*	miRNA (C19MC)	ISH	・*miR-517a/c* were more highly expressed in VT than in EVT.	7–8 w	2	[54]
				Functional analysis using EVT-like cells differentiated from iPSC	・When iPSC differentiated into EVT cells, *miR-517a/c* expression decreased and EMT-related gene expression increased.	NA	NA	
3	Takahashi (2017)	*miR-520c-3p*	miRNA (C19MC)	RT-PCR following LMD	・*miR-520c-3p* was highly expressed in the maternal decidua stroma.	7–9 w	5	[42]
				RT-PCR using human EVT cells isolated from 1st trimester placentas	・*miR-520c-3p* was more highly expressed in VT than in EVT.	7–11 w	6	
					・*miR-520c-3p* directly inhibited *CD44*.			
4	Xie (2014)	*miR-517-3p*	miRNA (C19MC)	RT-PCR comparison analysis (VT vs. EVT cells obtained from 1st trimester placentas by LMD)	・C19MC miRNAs were more highly expressed in VT than in EVT.	6–12 w	7	[43]
		*miR-518b*						
		*miR-519d*						
		*miR-520g*		Functional analysis using a human EVT cell line transfected with the C19MC cluster by a plasmid vector	・C19MC miRNAs attenuated the migration of an EVT cell line.	NA	NA	
		*miR-515-5p*						
		*miR-1323*						
5	Flor (2012)	*miR-520-3p miR-519a-3p miR-517a-3p*	miRNA (C19MC)	RT-PCR	・C19MC miRNAs were highly expressed in 1st trimester placentas.	10–14 w	4–20	[55]
				DNA methylation analysis	・The CpG island upstream of the C19MC cluster in the paternal allele escaped methylation.	NA	NA	
6	Hayder (2021)	*miR-210-3p*	miRNA	RT-PCR comparison analysis (among 1st trimester, 2nd trimester, preterm, and term placentas)	・The expression level of *miR-210-3p* decreased as gestational weeks increased.	1st: 5–12 w	8	[56]
						2nd: 13–25 w	10	
						preterm: 26–36 w	13	
						term: 37–40 w	17	
				Culture of EVT cells isolated from human 1st trimester placentas and a functional analysis	・*miR-210-3p* inhibited the migration and outgrowth of EVT cells.	6–9 w	10 in the miR-210-3p group 11 in the control group	
					・*miR-210-3p* directly inhibited *CDX*, a major transcription factor responsible for trophoblast self-renewal.			
7	Wang (2020)	*miR-210*	miRNA	ISH	・*miR-210* and its direct target *CPEB2* (HIF1α inhibitor) were co-located in SCT and VT.	7–8 w	NA	[57]
				Functional analysis using a human trophoblast cell line	・Positive feedback was shown between *miR-210* and HIFα mediated by *CPEB1*.	NA	NA	
8	Wang (2021)	*miR-210*	miRNA	ISH	・*miR-210* and *MEG3* were highly expressed in human 1st trimester trophoblasts. ・As VTs differentiated into EVTs and invaded the maternal decidua, *miR-210* expression decreased.	6–9 w	NA	[58]
		*MEG3*	lncRNA	Functional analysis using a human EVT cell line	・*miR-210* inhibited the EMT of an EVT cell line by suppressing *MEG3* expression.	NA	NA	
9	Gu (2013)	*miR-371-5p*	miRNA	Microarray analysis (1st vs. 3rd trimester placentas) ↓ Validation by RT-PCR and ISH	・The expression levels of *miR371-5p*, *miR-17-3p*, and *miR-708-5p* were significantly higher in 1st trimester placentas	6–8 w (1st) 30–38 w (3rd)	microarray: 6 in each group RT-PCR: 6 in each group ISH: 1 in each group	[59]
		*miR-17-3p*						
		*miR-708-5p*						
10	Singh (2017)	*miR-202-3p*	miRNA	Microarray analysis (placentas of women who later developed severe PE vs. those who did not ) ↓ Validation by RT-PCR	・Nine miRNAs were differentially expressed in the severe PE onset group at 11–13 w. ・The expression of miR-202-3p was 7-fold higher in the severe PE onset group at 11–13 w.	11–13 w	4 in each group	[60]
11	Wu (2016)	*miR-195*	miRNA	ISH	・*miR-195* and its direct target *ActR2B* were co-located in VT, SCT, and EVT.	7–9 w	NA	[61]
				Functional analysis using a human EVT cell line	・*miR-195* accelerated the invasion of an EVT cell line.	NA	NA	
12	Zeng (2020)	*H19*	lncRNA	ISH RT-PCR comparison analysis (normal vs. RM group)	・*H19* was located in trophoblasts, particularly EVT. ・The expression level of *H19* was significantly lower in the RM group.	6–12 w	ISH: 1 in each group RT-PCR: 12 in each group	[62]

C19MC: chromosome 19 microRNA cluster, EMT: epithelial-to-mesenchymal transition, EVT: extravillous trophoblast, iPSC: induced pluripotent stem cell, ISH: *in situ* hybridization, LMD: laser microdissection, lncRNA: long non-coding RNA, miRNA: microRNA, NA: not available, PE: preeclampsia, RM: recurrent miscarriage, RT-PCR: real time-polymerase chain reaction, SCT: syncytiotrophoblasts, VT: villous trophoblasts.

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
