# Peer review of "Non-Coding RNAs and Prediction of Preeclampsia in the First Trimester of Pregnancy"

_cells, 2022, doi:10.3390/cells11152428_

Round 1

Reviewer 1 Report

Pre-eclampsia is a syndrome clinically characterized by the combination of de novo hypertension during pregnancy and proteinuria.

As mentioned by the authors, placental dysfunction developed in the 1st trimester of pregnancy (incomplete invasion of the spiral arteries) is at the core of the pathology leading to preeclampsia, mainly preterm preeclampsia.

A growing body of scientific evidence suggests that multiple mechanisms may lead to this placental dysfunction. It is possible that new markers, including non-coding RNAs, may contribute to a better detection of preeclampsia.

However, upon reading this review, this assumption is speculative but interesting. I agree with the authors who conclude that there is a need for additional studies including analyses performed in early pregnancy. I do not have the knowledge and ability to evaluate the fundamental aspect of the results. However, I do not necessarily agree with the hypothesis that better detection between 14 and 16 weeks would necessarily be beneficial in predicting preeclampsia, when placental invasion seems to be predominantly present between 9 and 13 weeks of pregnancy.

The authors should therefore encourage research and analysis earlier than 11 weeks to be able to identify factors predisposing to placental dysfunction.

Author Response

Thank you very much for your comments on our manuscript.

Point 1

I do not necessarily agree with the hypothesis that better detection between 14 and 16 weeks would necessarily be beneficial in predicting preeclampsia, when placental invasion seems to be predominantly present between 9 and 13 weeks of pregnancy.

The authors should therefore encourage research and analysis earlier than 11 weeks to be able to identify factors predisposing to placental dysfunction.

Response 1

As you pointed out, considering the period of EVT invasion, we expect that validation using the samples from earlier in the 1st trimester will lead to the extraction of ncRNAs that are more strongly associated with early placentation and PE development. We have added this point in lines 752-756.

Reviewer 2 Report

This review article contains very interesting agenda suggesting ncRNA as possible new biomarkers for prediction of PE (especially for EOPE).

This article introduced a lot of ncRNA which could serve as possible predictive biomarkers for early/preterm onset PE. However, too many types of target ncRNA were mentioned in this article, and they did not show consistent results in the prediction of PE even when same ncRNA was used as biomarker. The diversity of target ncRNA makes it difficult to know which one could truly be a potential biomarker.

In order to use ncRNA as predictive markers, further research is still needed on what kind of sample should be used for analysis. (eg. miR-517-5p is upregulated in plasma who later developed PE, and miR-517-5p is downregulated in exosomes from plasma who later developed PE)

What we can say right now from this paper is that “‘some’ ncRNA might be associated with PE pathogenesis, and if further researches using this relationship between ncRNA and PE pathogenesis are conducted, ‘certain’ ncRNA could be used as predictive biomarkers for PE in the future”. It seems too early to mention ncRNA and prediction of PE together in the main title of this article.

Low-dose aspirin is very cheap medicine and does not cause serious complications when used during pregnancy. However, performing routine tests such as RT-PCR to measure ncRNA is very expensive. It should be further checked whether routine RT-PCR of ncRNA for PE prediction will be more cost-effective method than supplementing modified FMF prediction model and thereby expanding target group for low dose aspirin.

Author Response

Thank you very much for your comments on our manuscript.

Point 1

This article introduced a lot of ncRNA which could serve as possible predictive biomarkers for early/preterm onset PE. However, too many types of target ncRNA were mentioned in this article, and they did not show consistent results in the prediction of PE even when same ncRNA was used as biomarker.

What we can say right now from this paper is that “‘some’ ncRNA might be associated with PE pathogenesis, and if further researches using this relationship between ncRNA and PE pathogenesis are conducted, ‘certain’ ncRNA could be used as predictive biomarkers for PE in the future”. It seems too early to mention ncRNA and prediction of PE together in the main title of this article.

Response 1

As you pointed out, the ncRNAs shown in this review do not show the evidenced predictive ability for the onset of PE, but only show an association with PE. We described this point in lines 725-732.

Point 2

Low-dose aspirin is very cheap medicine and does not cause serious complications when used during pregnancy. However, performing routine tests such as RT-PCR to measure ncRNA is very expensive. It should be further checked whether routine RT-PCR of ncRNA for PE prediction will be more cost-effective method than supplementing modified FMF prediction model and thereby expanding target group for low dose aspirin.

Response 2

For the achievement of the ncRNA-based PE prediction model, the issue of cost is unavoidable. We have added this point in lines 756-762.
